# Central Venous Access: An Update on Modern Techniques to Avoid Complications

**DOI:** 10.3390/healthcare13101168

**Published:** 2025-05-16

**Authors:** Kai Woodfall, André van Zundert

**Affiliations:** 1Department of Anaesthesia and Perioperative Medicine, Royal Brisbane and Women’s Hospital, Brisbane, QLD 4006, Australia; k.woodfall@uq.edu.au; 2Royal Brisbane Clinical Unit, Faculty of Medicine, The University of Queensland, Brisbane, QLD 4029, Australia

**Keywords:** central venous catheter, anesthesia, complications, inadvertent arterial cannulation

## Abstract

**Background**: Central venous catheterization (CVC) is a frequently performed procedure in anesthesia and critical care settings. Modern procedures have improved significantly, particularly with increasingly sophisticated venous verification methods and ultrasound guidance. While the associated historical complication rates reflect this improvement, complications such as inadvertent arterial puncture, arterial cannulation, pneumothorax, deep vein thrombosis, and catheter-associated infection are still significant risks. **Methods**: This narrative review was constructed from a literature review using a search strategy of the MESH terms central venous access, central venous line, complications, insertion, and puncture, published between 2015 and 2025. Inclusion criteria included peer-reviewed full-text articles. Supplementary articles were included to construct the historical perspectives on central venous access and complications. **Results**: Our review offers a simple management algorithm for the mechanical complications of CVC insertion. This algorithm focuses on inadvertent arterial puncture/cannulation, with steps ranging from external compression to endovascular repair or surgical intervention. **Conclusions**: Moving forward, clinicians are encouraged to look into the future to predict what complications may arise as our modern patient cohort evolves. When complications develop, clinicians should know how to manage them to prevent further patient morbidity.

## 1. A Brief History of Central Venous Access

Central venous catheters (CVCs) are used daily in critical care to deliver medication, fluids, blood products, and hemodynamic monitoring. Up to 27 million CVCs are inserted worldwide annually [1]. This article reviews the insertion, complications, and management associated with central venous access, with a particular focus on inadvertent arterial injury.

This review was constructed primarily through a literature review from PubMed. The search strategy was designed in collaboration with The University of Queensland library staff, and the inclusion criteria were peer-reviewed full-text articles that included the MESH terms central venous access, central venous line, complications, insertion, and puncture and were published between 2015 and 2025. Additionally, supplementary articles on the historical aspects of central venous access and complications were found using simple search strategies on PubMed with no limit on publication date.

Central venous access has a rich history, shaping its modern techniques and complications. The first documentation of attempted central venous access occurred in 1733 by Englishman Stephen Hales (1677–1761), who measured a horse’s left internal jugular vein pressure using a glass tube (Figure 1) [2]. More than 100 years later, primitive techniques of central venous access were created, involving the great saphenous vein and brachial vein cut down and cannulation with long catheters to reach the superior vena cava. These techniques were time consuming, dangerous, and commonly associated with a high incidence of thrombotic complications [3].

Central venous catheters are routinely placed in one of three anatomical sites: the internal jugular (IJV), subclavian, or femoral vein [1]. The first documented cases of IJV cannulation using the landmark method occurred in 1969 [4]. Aubaniac first described subclavian cannulation in 1952 using an infraclavicular approach, while Yoffa described the supraclavicular approach in 1969 [5,6]. The US Army Surgical Research Unit documented one of the earliest descriptions of femoral vein cannulation in 1958. At that time, this technique was associated with significant complications, including lower limb edema, clotted tubing, and, most notably, thrombosis, often complicated by sheared-off polyethylene catheter tubing left in the venous system, which proved fatal for multiple patients. These complications hindered the mainstream adoption of the techniques [7]. The current landmark techniques for CVC insertion are detailed in Table 1.

Since the initial descriptions of modern central venous access, research has been conducted to improve the diagnosis and management of insertion-related complications. In 1985, Fabian and Jesudian described the novel technique of pressure manometry, which improved the safety of central venous cannulation by avoiding inadvertent arterial injuries [8]. This technique detected arterial punctures that could not be diagnosed through the traditional method of contrasting pulsatility and blood color.

Legler and Nugent first used ultrasound scan technology rather than traditional landmark methods for central venous access in 1984. This technology has profoundly reduced the incidence of mechanical complications during insertion due to its easier recognition of vessels and anatomical variations [3,9].

**Table 1 healthcare-13-01168-t001:** Overview of landmark techniques and common complications in central venous catheter insertion [1,10].

Choice of Vein	Anatomical Position	
**Internal Jugular Vein**		
Anatomy	Originating at jugular foramen, joins subclavian vein behind sternal extremity of clavicle.	
Approach	*Central Approach:* Insert needle 1 cm above apex of Sedillot’s Triangle *. Advance at 60 degrees to skin aiming at the ipsilateral nipple, blood should be obtained within 3 cm.	
	*Lateral/Posterior Approach:* Insert 2–3 cm above clavicle along posterior border of sternocleidomastoid (SCM). Direct needle towards jugular notch; blood should be obtained within 5 cm.	
**Subclavian Vein**		
Anatomy	The subclavian vein lies inferior to midpoint of clavicle and superior to the first rib. Joins internal jugular vein behind the sternal extremity of the clavicle.	
Approach	*Infraclavicular Approach:* Insert needle 1–2 cm inferior and lateral to the clavicular transition point. The needle should be aimed towards the sternal notch in a fashion parallel to the floor to avoid pneumothorax.	
	*Supraclavicular Approach:* Insert needle 1 cm superior and lateral to the junction of the lateral border of the clavicular head of SCM and the clavicle. The needle is oriented 5–15 degrees posteriorly off a coronal plane along a line that would bisect the angle of the clavicle and sternocleidomastoid.	
**Femoral Vein**		
Anatomy	The femoral vein runs medially to the femoral artery in the femoral triangle, becoming the external iliac vein as it passes the inguinal ligament.	
Approach	Insert needle 1 cm medial to the point of maximal femoral arterial pulsation in the anterior superior thigh within the femoral triangle. Insert at a 45-degree angle along the same line as the arterial pulsation.	
Complications	The common complications for all insertion sites are catheter misplacement/failure, arterial puncture/cannulation, pneumothorax, catheter-associated infection, and deep vein thrombosis. It should be noted that there is a negligible risk of pneumothorax with CVC insertion into a femoral vein.	

* Sedillots triangle is defined by the posterior border of the sternal head of the SCM, the anterior border of the clavicular head of the SCM, and the superior aspect of the clavicle inferiorly.

## 2. The Modern Procedure

The modern approach to gaining central venous access has improved significantly since the previously mentioned historical methods. In 2020, the American Society of Anesthesiologists published updated practical guidelines for achieving central venous access and preventing complications. This guideline recommends using aseptic techniques (e.g., hand washing) and maximal barrier precautions (e.g., sterile gowns, gloves, caps, and masks that cover both the mouth and nose, along with full-body patient drapes). Furthermore, experts suggest utilizing chlorhexidine-based skin cleansing solutions or povidone-iodine solutions when those are contraindicated [11].

When determining the appropriate type of central line for placement, guidelines recommend adhering to these principles: (1) keep the number of lumens to a minimum; in standard practice, often three lumen CVCs are used; (2) if a lipid-containing solution is used, for example, in the administration of total parenteral nutrition, ensure that it has a dedicated lumen, and (3) consider how long the CVC line needs to remain in situ [11]. Patients at high risk of developing catheter-associated infections include those who will have catheters in for more than 7 days or those with specific risk factors, such as burns or immunocompromised conditions. For these patients, an antimicrobial-coated line should be considered [12]. While various CVC sizes, types, and introducer needles are available, current research and expert recommendations indicate that catheter size and type should be selected based on local protocols and operator experience. No evidence demonstrates a clinical difference in outcomes among the various options. Additionally, clinicians should utilize the smallest catheter necessary for the clinical situation [11].

When deciding between right- and left-sided options for inserting CVCs, the current literature and expert opinion suggest prioritizing right-sided insertion sites for internal jugular vein catheterization. However, both options should be considered depending on clinical circumstances [13]. These studies found that, when comparing the right and left internal jugular veins (RIJV and LIJV), the RIJV is larger and less likely to be positioned anteriorly relative to the common carotid artery, thus minimizing the risk of a ‘through and through’ inadvertent arterial puncture [14,15,16]. As for subclavian or femoral cannulation sites, expert consensus suggests scanning both sides and choosing whichever the operator feels most comfortable with. A 2024 RCT suggests that the left subclavian vein has a straighter anatomical path to the SVC and found that there is a lower rate of catheter malposition in left-sided subclavian insertions when compared with the right side (4.5% vs. 13.8%, *p* = 0.001) [17].

Two mainstream insertion techniques are commonly used: the wire-through-thin-wall needle technique (Seldinger technique) and the catheter-over-the-needle-then-wire-through-the-catheter technique (modified Seldinger technique). When inserting into the internal jugular or femoral veins, operators should choose the method with which they feel most experienced. However, when inserting into the subclavian vein, some RCTs suggest that a thin-walled technique (Seldinger technique) results in a lower complication rate than a catheter-over-needle technique (Modified Seldinger) [18,19].

For CVC insertion, an ultrasound with a sterile probe cover and sterile gel is recommended to locate the relevant venous and arterial anatomy. The needle is inserted under ultrasound guidance into the central vein, and blood flow should return through the needle, indicating intravascular access. In the Seldinger technique, a guidewire is inserted through the introducer needle into the vein, and its position is checked with ultrasound. The modified Seldinger technique feeds the cannula into the vein, and the guide wire is inserted through the cannula under ultrasound guidance. The guidewire is then used to insert the required dilator or multi-lumened CVC into the vein [11].

Verification methods for correct needle, wire, and catheter placement in the venous system include ultrasound, manometry, waveform analysis, fluoroscopy, venous blood gas, echocardiography, transesophageal echocardiography, and chest radiography (CXR) [11]. Figure 2 shows how these verification methods can help ascertain a CVC’s venous placement [20].

Historically, post-procedural CXR has been the modality of choice for determining the CVC position and development of iatrogenic complications [11]. The current guidelines recommend that the optimal tip position is at the junction of the lower superior vena cava (SVC) and the right upper atrium when viewed on CXR [21]. This allows high blood flow to prevent thrombosis while also lying outside the right atrium to avoid arrhythmias from catheter tip irritation of the right atrial wall. The surface landmarks correlating to this position for measuring catheter length are from the intended puncture site to the angle of Louis, i.e., the manubrio-sternal joint [10]. An MRI study of catheter tip positions compared to CXR radiographic landmarks showed that the right tracheobronchial angle is the most reliable landmark for the upper SVC. It showed that catheter tips placed caudad to this landmark by no more than 2.9 cm, resulted in the best positioning outcomes within the SVC [22]. The carina is another reliable radiographic landmark that can be used to place CVCs. In 2000, a study published in the *British Journal of Anaesthesia* showed that the carina is reliably visible in poor-quality CXRs, and catheter tips positioned above this landmark on CXR minimize the chance of potentially fatal cardiac tamponade from deep CVC insertion [23]. In 2019, the Royal Brisbane and Women’s Hospital promoted an insertion bundle checklist in which clinicians judged the acceptable tip position, which was the upper, mid, or lower SVC and the cavo-atrial junction, from the post-procedure CXR. The quality improvement project utilized similar radiographic landmarks to assess the optimal tip position. This audit showed that more CVCs were inserted at an appropriate depth, reducing the number of ‘too high’ CVCs [24].

However, the recent literature has shown that dynamic ultrasound scanning is non-inferior to CXR for evaluating post-procedure positioning. A 2017 meta-analysis by Ablordeppy et al. shows that bedside ultrasound and TTE can help determine the final catheter position. It reported a pooled positive likelihood ratio of 31.12, indicating that ultrasound is sufficient to confirm CVC malposition, but a negative likelihood ratio of 0.25 suggests that it is insufficient to rule out catheter malposition [25]. Therefore, clinicians could feasibly use bedside ultrasound to decide if post-procedure CXR is necessary when there is clinical suspicion of CVC malposition. A 2018 meta-analysis found that the diagnostic accuracy of ultrasound for detecting CVC malposition has a pooled specificity and sensitivity of 98.9 (95% CI: 97.8–99.5) and 68.2 (95% CI: 54.4–79.4), respectively. It was also a more reliable method for detecting immediate mechanical complications of CVC insertion, such as pneumothorax or inadvertent arterial puncture. These data suggest that the CVC position is best verified by combining vascular ultrasound for guidewire and cannula position in the vessel with transthoracic echocardiography (TTE) for the CVC tip position [26].

While our review focuses on non-tunneled conventional CVCs for access to the central venous system, it is important to note that many other options are available for accessing the central venous system. Although an extended review of these options falls outside the scope of this paper, as critical care providers rarely perform their insertion, they will be briefly discussed here.

Peripherally inserted central catheters (PICCs) represent a type of non-tunneled central venous access. Long catheters are inserted into peripheral veins so that their tips terminate in the distal one-third of the superior vena cava, similar to traditional CVCs. They are associated with complications, such as central line-associated bloodstream infections, catheter occlusions or failure, and deep vein thromboses (DVTs) [27,28]. Data from a 2021 multicenter study suggest that PICC-associated bloodstream infection rates are as high as 1.6%, while catheter occlusion rates can reach up to 7.0% [27] The incidence of PICC migration or dislodgement is between 5% and 31% [29] Depending on the patient cohorts examined, the rates for deep vein thrombosis secondary to PICCs vary widely, ranging from 0.7% to 91% in the most critically ill patients [27,29].

Tunneled central venous access devices (CVADs) provide an alternative form of access and include Hickman lines, Broviac lines, or totally implantable ports. These devices are suitable for long-term medication administration and are selected when treatment is likely to last months to years. They can be inserted under the skin percutaneously or through a surgical cut-down using local or general anesthesia [30]. The choice of these devices is often patient-specific and depends on their intended use [31,32]. Common complications associated with these long-term CVADs include catheter blockage, catheter-associated infection, infection of the exit site or tunnel, and deep vein thrombosis [30].

## 3. Complications

Despite their frequent use and improvements in procedure, the immediate insertion process poses various risks to the patient, including arterial puncture or cannulation, pneumothorax, catheter mispositioning, a bleeding event requiring action, nerve injury, arrhythmia, and cardiac tamponade. Furthermore, delayed complications can result from the ongoing use of CVCs, including catheter malfunction, deep vein thrombosis, and catheter-associated infection [1,26]. Table 2 documents the current complication rate for both immediate and delayed complications of CVC insertion. Case reports also describe a variety of rare complications, including mediastinal hematoma, the loss of guidewire, the intracardial entrapment of pulmonary artery catheters, and massive hydrothorax [33,34,35,36,37,38].

**Table 2 healthcare-13-01168-t002:** Common immediate and delayed complication rates [1].

**Immediate Complication**	**Total Catheters Analyzed**	**Complication % (95%CrI)**
Placement failure	17,407	2.04% (1.09–3.44)
Arterial puncture	22,296	1.62% (1.15–2.20)
Arterial cannulation	6489	0.28% (0.01–1.00)
Pneumothorax	32,665	0.44% (0.27–0.65)
**Delayed Complication**	**Total Catheter Days Analyzed**	**Complication % per Catheter Day (95%CrI)**
Deep vein thrombosis	73,894	0.27% (0.1–0.62)
Catheter-associated infection	549,246	0.48% (0.34–0.66)

95% CrI indicates 95% credible interval.

These risks depend on the chosen location of the catheter, operator experience, adherence to sterility measures, and whether ultrasound is used during the insertion and confirmation process [1,11,39,40]. Misplacement of the catheter is most commonly documented, occurring between 10.9 and 34.4 events per 1000 catheters inserted [1]. However, these rates significantly decrease to as low as 5.2 events per 1000 when dynamic ultrasound is utilized during insertion [1]. It should also be noted that the data for Table 2 are drawn from a recent meta-analysis combining landmark methods and dynamic ultrasound scan-assisted CVC insertions. Notably, the incidence rate of arterial puncture using the landmark method has been reported to be as high as 9%. In comparison, the rate of arterial cannulation with the landmark method is reported to be as high as 0.5% [41,42,43]. The difference in incidence can be attributed to cases where arterial puncture was noted, and appropriate measures were taken to prevent arterial cannulation.

The historical incidence of complications has decreased over time due to procedure improvements, as shown in Table 3. A recent meta-analysis on central venous catheterization published by Teja et al. drew data from 2015 to 2023 and demonstrated the lowest complication rates in all areas compared to historical data [1]. In 2015, the *New England Journal of Medicine* published a multicenter randomized controlled trial conducted in nine hospitals in France from December 2011 through June 2014 [44]. They analyzed the complication rate of central venous catheters using three different insertion sites. Regarding site-specific differences, they concluded that subclavian vein catheterization was associated with a lower risk of catheter-associated infections and deep vein thrombosis but a higher risk of pneumothorax compared to internal jugular or femoral vein catheterization. From a historical comparison perspective, overall data reported that symptomatic deep vein thrombosis and catheter-associated infections occurred in 0.5% to 14% and 0.5% to 1.2% of central venous insertions, respectively [44]. Their data showed immediate mechanical complication rates of 0.7% to 2.1% and defined major mechanical complications (grade 3 or higher) per the modified National Cancer Institute Common Terminology Criteria for Adverse Events, version 4.0 [44]. These included the pneumothorax requiring any operative management or leading to clinical symptoms for the patient, arterial or venous punctures resulting in severe symptoms or requiring operative intervention, or any bleeding event necessitating the transfusion of RBCs or operative management [44].

**Table 3 healthcare-13-01168-t003:** Overview of historical complication rates.

Immediate Complications	2024 (95% CrI) [1]	2015 [44]	1997 [45]	1985 [46]	1971 [47,48]
Placement failure	2.04% (1.09–3.44)	x	(1.9–5.2%)	x	6.5%
Arterial puncture	1.62% (1.15–2.20)	x	5.20%	3.8%	19%
Pneumothorax	0.44% (0.27–0.65)	x	0.50%	2.3%	6%
Major mechanical complication *	x	0.7–2.1%	x	x	x
**Delayed Complications**					
Deep vein thrombosis	0.27% (0.1–0.62)	0.5–1.4%	x	x	x
Infection	0.48% (0.34–0.66)	0.5–1.2%	x	5.7%	3.5%

95% CrI indicates 95% credible interval. x indicates no data was available for this specific complication in this year. * Major mechanical complication defined as any of the following: (1) pneumothorax requiring any operative management or that led to clinical symptoms for the patient; (2) arterial or venous punctures leading to severe symptoms or requiring operative intervention; (3) bleeding event requiring transfusion of RBCs or operative management.

In 1997, Yilmazlar et al. published a prospective study of 1303 central venous catheters (CVCs) and compared their incidence of complications with other available studies at the time. Their research revealed a 5.2% incidence of arterial puncture (4.1–11.3% reported in other studies) and a 0.5% (0–5.2% in other studies) incidence of pneumothorax. They noted that rates of catheter malposition in the literature from that period ranged from 1.9% to 5.2%. The authors observed that, compared with other studies available at the time, the lower incidence of pneumothorax was likely due to their choice of the internal jugular vein (IJV) as the cannulation site [45]. A prospective study published in 1985 by Sitzmann et al. followed the historical trend of increasing complication rates [46]. These data examined 200 consecutive patients at Johns Hopkins Hospital requiring central lines for total parenteral nutrition. The rates of immediate complications are displayed in Table 3, and it should be noted that catheter-associated infection rates are likely skewed due to the administration of total parenteral nutrition. One of the earliest studies examining the complication rates associated with central venous catheterization was a two-part prospective study by Bernard and Stahl published in 1971. This research focused solely on subclavian vein catheterization. The rates of placement failure, pneumothorax, arterial puncture, and infection were significantly higher than in more recent data [47,48].

These data show a linear improvement in complication rates over time, with noticeable improvement once the development of ultrasound-guided techniques became mainstream for critical care providers post-1984. As shown in Table 4, evidence shows that dynamic ultrasound markedly reduces complications associated with central venous access compared to traditional landmark techniques. Apart from misplacing the CVC, arterial puncture with the introducer needle is the most common significant complication. Furthermore, of the immediate mechanical complications, arterial cannulation has the highest potential morbidity and mortality per incidence [49]. A 2023 educational article published in the *British Journal of Anaesthesia* provides guidelines (Figure 3) aimed at preventing mechanical complications associated with CVC insertion, effectively summarizing some of the modern key points. It contends, however, that no single technique is sufficient to avoid all complications related to CVC insertion [20].

**Table 4 healthcare-13-01168-t004:** Complication rates of CVC insertion: Dynamic ultrasonography versus the landmark method.

Complication [1]	Ultrasonography	Landmark Method
Placement failure	1.19% (0.51–2.46)	2.90% (0.75–9.63)
Arterial puncture	1.35% (0.91–1.92)	6.88% (3.24–13.75)
Arterial cannulation	n/a	n/a
Pneumothorax	0.24% (0.12–0.43)	0.99% (0.33–2.62)
**Delayed complications [1]**		
Deep vein thrombosis	0.12% (0.02–0.67)	n/a
Infections	0.42% (0.23–0.72)	2.18% (0.29–15.4)
**Other complications * [39,40]**		
Internal jugular vein	8 per 1000 (3–17)	23 per 1000
Subclavian vein	30 per 1000 (7–127)	105 per 1000
Femoral vein	6 per 1000 (1–28)	13 per 1000

n/a—nil sufficient data for adequate analysis. * Other complications defined as the following: thrombosis, embolism, hematomediastinum and hydromediastinum, hematothorax and hydrothorax, pneumothorax, subcutaneous emphysema, and nerve injury.

## 4. Inadvertent Arterial Puncture

Unintentional cannulation of an arterial vessel with a dilator or large-bore catheter is a dangerous and difficult-to-manage complication for the modern anesthesiologist and intensivist. The American Society of Anesthesiologists Closed Claims Project database suggests that, since 1990, the majority of immediate complications associated with CVCs leading to insurance claims have been vascular injuries [50].

The modern literature shows that the pooled rate of arterial cannulation was 2.8 events per 1000 catheters placed (Table 2) [1]. Table 4 and Table 5 show that incidence rates were significantly affected by the insertion location and whether ultrasound was used in the insertion process.

The complications associated with arterial puncture or cannulation are severe and include stroke, pseudoaneurysm, airway obstruction, further surgery for vascular repair, or death [49]. Commonly, the methods for avoiding the arterial placement of a CVC were to observe the color and pulsatility of the blood from the introducer needle before placing a guidewire or to test the degree of oxygenation of the blood with sampling [51]. However, these methods can be unreliable due to systemic hypotension, arterial desaturation, catheter kinking, or occlusion [51]. In 1983, a retrospective cohort study by Jobes et al. of over 1200 attempts at IJV access reported that, on clinician assessment of color and pulsatility, 0.8% of needle punctures were incorrectly identified as being in the venous system but could be correctly identified as arterial on pressure waveform analysis [52,53]. Figure 4 shows an example of the difference between arterial and venous waveforms with an ECG for comparison.

**Table 5 healthcare-13-01168-t005:** Inadvertent arterial injury by CVC insertion site [1].

Location	Total Catheters Analyzed	Complication % (95%CrI)
Internal Jugular Vein	8852	1.19% (0.5–2.47)
Femoral Vein	1200	2.77% (0.25–22.58)
Subclavian Vein	4253	3.59% (1.32–8.61)

95% CrI indicates 95% credible interval.

Furthermore, Ezaru et al. published a retrospective study in 2009 of over 9000 CVC insertions [51]. They determined that an inadvertent arterial puncture could be correctly identified in virtually all cases using pressure manometry combined with an assessment of color and pulsatility. In contrast, again, with reliance on color and pulsatility alone, 0.8% of insertions would have led to arterial cannulation. As previously discussed, the current standard of practice is the real-time use of ultrasound to confirm the location of the needle, guidewire, and catheter tip.

## 5. Management of Inadvertent Arterial Puncture/Cannulation

When an uncomplicated arterial puncture with an introducer needle is identified, the most common management is applying external pressure to prevent hemorrhagic complications, which is often effective. However, for select groups of patients with bleeding disorders, difficult-to-compress vascular punctures (e.g., subclavian artery), or extensive atherosclerotic disease, postponing any subsequent operation and ongoing neurological follow-up could be recommended [49,54].

However, clinicians must know the management options when arterial punctures are not identified and inadvertent arterial cannulation occurs. A 2009 case series by Blaivas reported six inadvertent arterial cannulations that occurred despite the use of best practice ultrasound [55]. This review notes that the assumption that “serious complications no longer arise when ultrasound is used” is incorrect. Table 6 summarizes these cases, including the outcome for each patient. This confirms that no single technique can prevent inadvertent cannulations of arteries during CVC placement. Historically, attempts to remove the dilator or inadvertently intra-arterially placed CVCs and apply direct pressure (the “pull and pressure” approach) have led to significant morbidity and mortality for patients. The ideal management strategy for the modern clinician is a combination of CT angiogram, direct angiography, and endovascular closure techniques, which have been successful with minimal complications in various case reports [56,57,58,59]. Given the minimally invasive nature of angiography compared to open surgical options and similar clinical outcomes, it should be considered the first-line option for managing these complications.

**Table 6 healthcare-13-01168-t006:** Video analysis of six accidental arterial cannulations during CVC insertion using dynamic ultrasound guidance [55].

Age	Mechanism of Injury	Outcome
67	Attending emergency physician and resident inserted CVC into an awake septic patient with multilobar pneumonia. The needle went through IJV into carotid artery.	Patient died after failed emergency intubation for worsening respiratory distress from mediastinal hematoma.
75	Attending physician inserted into awake patient with urosepsis. The catheter went through femoral vein into femoral artery.	Vascular surgery for arteriovenous fistula.
48	The needle inserted into an intubated patient with pulmonary edema went through a collapsed IJV and entered carotid artery posterior to the IJV.	Surgery for tear and focal dissection of carotid artery.
67	An awake oncology patient with esophageal cancer had a CVC inserted for pulmonary edema. Guidewire traveled through IJV and its posterior wall and into the carotid artery.	Hematoma with respiratory distress requiring emergent intubation.
69	An awake patient with respiratory distress had a CVC inserted. The needle penetrated the carotid artery, which was very close to the IJV.	Emergency carotid artery repair; the patient died of complications.
14	An awake patient with likely urosepsis had a CVC inserted by an emergency physician. The needle penetrated the rear wall of IJV and entered carotid artery.	Central catheter removed; bleeding eventually stopped.

In 2008, Guilbert et al. published a retrospective analysis and literature review that showed removing the malpositioned catheter and applying direct pressure led to significant complications in 47% of patients, including stroke or death. Their results showed the immediate stroke risk following a carotid arterial CVC removal was 5.6% and thus recommend prompt imaging and neurological assessment, which may necessitate the postponing of elective surgeries in anesthetized patients. Their data showed that open surgical repair or endovascular intervention was not associated with increased morbidity or mortality. Particularly, endovascular repair was recommended for sites of arterial trauma below the clavicle [54].

A 2017 systematic review of the management of inadvertent arterial cannulation compared the treatment options of the three treatment modalities. They found that complications arose in 94.4% of arterial cannulations managed with external compression, notably hematoma formation, hemorrhage, and embolic events. Significantly, 16.7% of these patients died due to embolic stroke. Comparatively, in the 37 cases of endovascular treatment, there were only two complications (5.4%), with one case of inability to stop the hemorrhage and one embolic stroke. In 37 cases of open surgical management, no complications were reported. They concluded that, while surgery appears to be more optimum in terms of closure complications, it is more invasive and may carry more morbidity due to the open nature of repair compared to endovascular methods. They contended that it might also be less favorable in subclavian artery repairs, as this area is more difficult to access and includes the risk of a general anesthetic in a critically unwell patient. They believe clinicians should consider both angiography and surgery depending on the individual patient and location factors [60].

A 2022 case series by Papastratigakis et al. analyzed three more modern cases of inadvertent arterial cannulation and again recommended against removing the catheter and direct pressure, which led to complications like hematoma, airway obstruction, and stroke [43]. Their report encouraged that the current gold standard of care is that, once inadvertent arterial cannulation is identified, clinicians should discuss an angiographic approach to management with an interventional radiologist. Immediate consultation with a general or vascular surgeon is also appropriate, depending on the clinical setting [43].

Much of the current literature suggests the consideration of prophylactic anticoagulation due to the prevalence of thrombotic complications after arterial cannulation, leading to embolic stroke. Some authors hypothesize that a thrombus forms around the catheter, which dislodges and embolizes during catheter removal [61]. While there is no consensus on the requirement for anticoagulation, some vascular protocols suggest full heparinization and ultrasound evaluation of the injured artery (particularly carotid arteries) to look for thrombus formation or atherosclerotic calcifications and back bleeding of the catheter before removal. Furthermore, they recommend considering open surgical techniques over endovascular closure in cases with >4 h before the arterial cannulation is identified [60,61]. Figure 5 presents our recommended management algorithm for contemporary practitioners. If arterial cannulation occurs with the dilator or catheter, endovascular or angiographic closure techniques are the gold standard of care. However, in clinical situations lacking access to angiography or endovascular closure, open surgical repair should be considered for patients stable enough to tolerate the procedure, which often requires general anesthesia [60]. If these patients are not stable enough to tolerate open surgical repair, consider immediate consultation with interventional radiology and vascular surgery to arrange potential transfer to a facility that can offer endovascular closure. In practice, we assert that proceduralists, interventional radiologists, general surgeons, and vascular surgeons should all be engaged, if available, in a shared decision-making process to determine the most appropriate intervention for each patient.

## 6. Pneumothorax

The development of a pneumothorax through the process of obtaining central venous access has been consistently documented historically as a common complication. The 2024 meta-analysis by Teja et al. suggests that pneumothorax rates range from 0.3 to 54.9 events per 1000 catheters placed, reflecting substantial between-study heterogeneity and a notable negligible risk of pneumothorax when studies involving femoral vein catheter placement were analyzed [1].

The previous standard investigation for the diagnosis of a pneumothorax is CXR. However, some international clinical guidelines note that immediate post-procedure CXR is often insufficient, as this complication develops insidiously [62]. For the modern clinician, a recent meta-analysis reports that the sensitivity and specificity of ultrasound to detect post-procedure pneumothorax are nearly 100%, and this has been reflected consistently in the literature [25,63,64]. The first documentation of ultrasound for the diagnosis of pneumothorax occurred in 1986 by a veterinarian in a study of horses, and it has since been shown that it is superior in efficacy and speed to CXR in the diagnosis of pneumothorax in the supine position and critically ill patients [25,65].

The current recommended management of pneumothorax by the American Society of Anesthesiologists’ 2020 report is the insertion of a chest tube if a pneumothorax develops [11]. The Swedish Society of Anaesthesiology and Intensive Care Medicine’s 2014 guidelines elaborate further and recommend that, in cases where patients do not develop clinical signs of dyspnea, tachypnoea, decreased peripheral oxygen saturations, or cough, and the pneumothorax corresponds to less than 30% of the pleural cavity, drainage is not required [62].

## 7. Other Complications

Table 7 summarizes the common complications reported by the ASA and their diagnosis and management. Given the invasive nature of central venous catheterization and its frequency in modern medicine, various rare complications are intermittently reported in case studies. We propose that modern clinicians be aware of these risks despite their small incidence rate.

**Table 7 healthcare-13-01168-t007:** Common complications of CVC placement as reported by ASA 2020 report—brief diagnosis and management summary [11].

Immediate Complications	Diagnosis	Management
Placement Failure	Historically, CXR has been the gold standard. Ultrasound (USS) and TTE are non-inferior and faster in recent meta-analyses.	Remove misplaced catheter.
Arterial Puncture	Combination blood color and pulsatility assessment, pressure manometry, and dynamic USS for needle tip position.	Often, no invasive management is required apart from external compression if it is identified before further cannulation.
Arterial Cannulation	Combination of blood color and pulsatility assessment, pressure manometry + waveform analysis, and dynamic USS for catheter position.	Leave the catheter in situ, and follow the pathway documented in Figure 5.
Pneumothorax	Bedside ultrasound is faster, more sensitive, and more specific than CXR for identifying pneumothorax after central venous catheter insertion.	Patients with large pneumothoraces >30% or respiratory distress may benefit from the insertion of a chest drain.
**Delayed Complications**		
Deep Vein Thrombosis	Combination of clinical assessment, D-dimer, and USS.	The catheter may be left in situ. Anticoagulation for as long as catheter is in situ or ≥3 months, whichever is longer.
Catheter-Associated Infection	Clinical assessment and blood cultures.	Remove the catheter and send for tip culture. Antibiotics per local guidelines. Prophylaxis is not recommended

### 7.1. Retained Guidewire

While an adequate CVC insertion technique should prevent this complication, multiple case reports have documented the complications that arise after the guidewire’s vascular retention during insertion. These case reports document that unrecognized retained guidewires lead to complications, including embolization, fragmentation, infection, arrhythmias, cardiac perforation, stroke, and migration through soft tissue [33,66,67,68,69,70]. Current international guidelines recommend that, if the complete guidewire is not found in the procedural field at the end of the insertion procedure, a CXR should be ordered immediately to determine if it is retained in the vascular system. In this scenario, immediate consultation with vascular surgeons or interventional radiologists should occur [11].

### 7.2. Hemothorax/Mediastinal Hematoma/Cardiac Tamponade

Hemothoraces or mediastinal bleeding can occur during central venous catheterization if an arterial or venous perforation communicates with the space. The potential pleural space is large, up to approximately 3 L, as the lung is completely compressible. If a hemothorax develops post-insertion, the consensus recommendations are to leave the catheter in situ, consider a chest drain, serial hematocrit measurements, volume replacement, and urgent vascular surgical consultation [11,49].

Mediastinal hematoma case reports usually document the insidious hematoma development after the insertion procedure. The mediastinal space is much smaller than the pleural space, so this complication leads to airway manifestations, including dyspnea, decreased peripheral oxygen saturations, and potential airway compression [34,37]. If this complication is suspected, the immediate priority is controlling hemorrhage from the puncture site and ongoing monitoring. Even large hematomas can usually be managed conservatively [71].

Cardiac tamponade is a life-threatening complication associated with central venous access attempts. This complication occurs more frequently in neonatal central venous cannulation, and its incidence is not well established in the literature, ranging from 0.01% to 0.3%, but it has a reported mortality ranging from 30% to 100% if there is associated ventricular perforation [72,73].

### 7.3. Chylothorax/Infusothorax/Hydrothorax

Rare complications of non-blood-containing fluid entering the mediastinum or pleural cavity because of CVC insertion have been documented in sporadic case reports. Infusothorax/hydrothorax mechanisms have been proposed to include the erosion of the subclavian or internal jugular vein wall, leading to fluid infusion into the mediastinum or pleural cavity [74]. These cases have been managed primarily with the cessation of the infusion and insertion of a chest drain if the patient is systemically unwell [74,75].

Chylothorax development has various attributable causes, including the insertion of CVCs. Iatrogenic injury to the lymphatic system leads to the accumulation of chyle in the pleural cavities, which can present as shortness of breath, cough, or chest discomfort [76]. While no randomized control trials are available to guide management, recent publications suggest that conservative measures such as low-fat diets and/or chest tube placement are sufficient for managing low-output chylothoraxes. For patients who do not respond to conservative management or have high-output chylothoraxes (>1000 mL per day), surgical intervention is warranted [76].

### 7.4. Delayed Complications

Table 2 and Table 4 detail the incidence rates of deep vein thrombosis associated with CVC and catheter-associated infections. Interestingly, using ultrasound scanning in the insertion process potentially reduces the risk of both events despite not being an immediate insertion-related complication [1,39,40].

Deep vein thrombosis associated with CVC can be diagnosed with clinical decision scoring (Constans score), D-dimer testing, and ultrasound imaging [77]. Upper limb catheter-associated deep vein thrombosis often presents as asymptomatic, inability to flush or aspirate from the catheter, or catheter-associated infection [78]. The current recommendations from international committees for treatment are to consider anticoagulation for as long as the catheter remains in situ or for ≥3 months, whichever is longer. It should be noted that these guidelines note there is a lack of strong evidence for the duration of anticoagulation after catheter removal, and all decisions should be made on a case-by-case basis [79].

Catheter-associated infections are commonly reported. Notably, despite modern advances in aseptic techniques, over 80,000 CVC-associated bloodstream infections occur annually in the United States [80]. If an infection is suspected, international guidelines recommend that blood cultures be taken from both the catheter and a peripheral site, and the catheter tip should be sent for culture [62,81]. Antimicrobial therapy for suspected catheter-associated infections should empirically target the likely causative organisms while awaiting culture sensitivities. The common causative organisms for these infections include Staphylococcus aureus and coagulase-negative staphylococci. Candida species and Gram-negative bacteria are also prevalent in immunocompromised patients [81,82]. Given the regional differences in antimicrobial resistance, infections should be treated with appropriate antimicrobials according to local guidelines. However, in general, these infections can be treated empirically with vancomycin, with the addition of an aminoglycoside if there is a suspected high risk of Gram-negative infection [81]. The American Society of Anesthesiologists and the Centers for Disease Control do not recommend routine IV antibiotic prophylaxis [11,83].

### 7.5. Future Perspectives

Given the history of central venous cannulation, our modern advances in techniques and equipment have been instrumental in reducing the risks associated with this critical procedure. As such, modern clinicians should begin to preempt the possible future complications that a contemporary patient cohort may encounter. Looking ahead, obesity has been identified through meta-analysis as an independent risk factor for difficult peripheral intravenous cannulation [84]. As modern society progressively increases its average BMI, and, in particular, the prevalence of extreme obesity, research has begun to focus on ensuring that CVC remains a safe procedure for these patients [85]. This includes head and neck positioning strategies, passive leg raising, and other patient positions to enhance central venous accessibility under ultrasound guidance [86,87,88].

Further research could consider the complications of increased body mass and subcutaneous tissue. Hypothetical new complications requiring further investigation could involve subcutaneous neck adiposity weighing down upper body CVCs, leading to post-verification placement failure, inadvertent arterial puncture due to the subcutaneous weight forcing catheters through posterior venous vessel walls and into adjacent arterial walls, or catheter failure from kinking under the weight. We must remain prepared for new complications as we advance techniques to provide central venous access.

## 8. Conclusions

Central venous catheterization remains one of the most critical procedures in anesthesia and critical care medicine. The transition from invasive methods, such as the great saphenous vein cut-down, to ultrasound-guided techniques has significantly improved patient safety and physicians’ capability to manage critically ill patients. Since the 1980s, ultrasound guidance has revolutionized vascular access, enabling real-time vessel visualization and decreasing complications. However, no single technique is adequate to prevent all complications related to CVC insertion. Immediate complications such as arterial puncture, arterial cannulation, and pneumothorax/hemothorax remain prevalent. Clinicians must be prepared to tackle these challenges, using ultrasound and additional tools like pressure manometry and waveform analysis as required. Research indicates that ultrasound guidance can significantly reduce the risk of arterial puncture, pneumothorax, and catheter misplacement compared to traditional landmark approaches. In addition to attempting to prevent the development of complications with modern techniques, clinicians should be aware of the management pathways if they arise. This review offers a suggested management pathway for the most common complications, particularly the management of inadvertent arterial cannulation. Clinicians should remain current with the latest endovascular techniques and indications to prevent poor outcomes for patients who undergo inadvertent arterial cannulation. Future research should examine how to best support patients undergoing angiographic and endovascular closure techniques after arterial cannulation and the need for ultrasound evaluation and anticoagulation in these cases. Additionally, increasing patient obesity changes optimal CVC insertion and associated risks, and more data are needed to develop evidence-based guidelines and management in this cohort.

## Figures and Tables

**Figure 1 healthcare-13-01168-f001:**
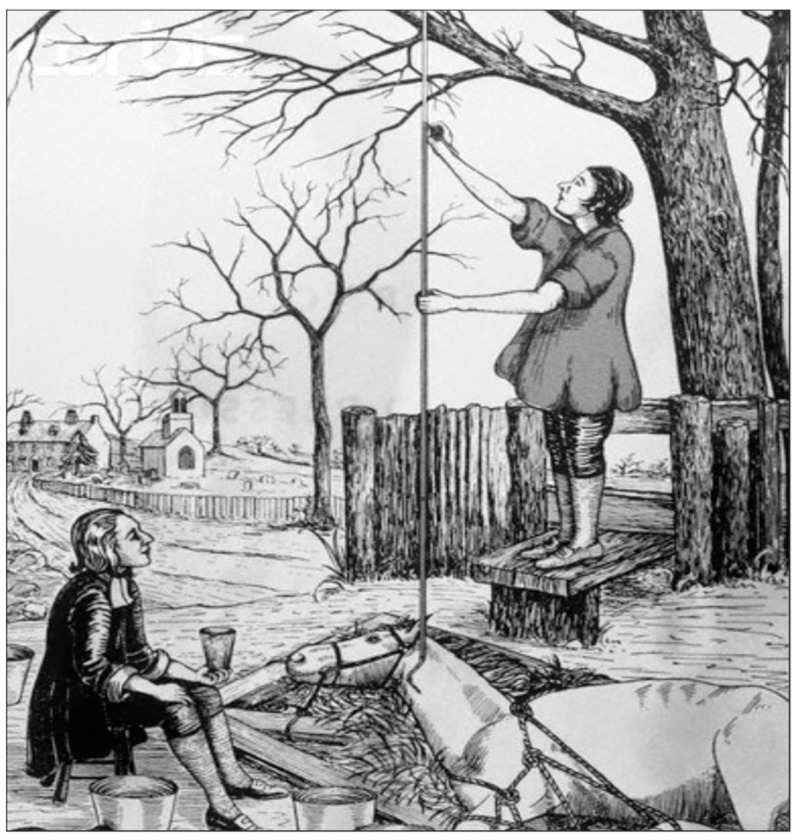
The first catheterization of a central vein performed by Stephen Hales in 1733 (with permission of Sette et al. 2012) [2].

**Figure 2 healthcare-13-01168-f002:**
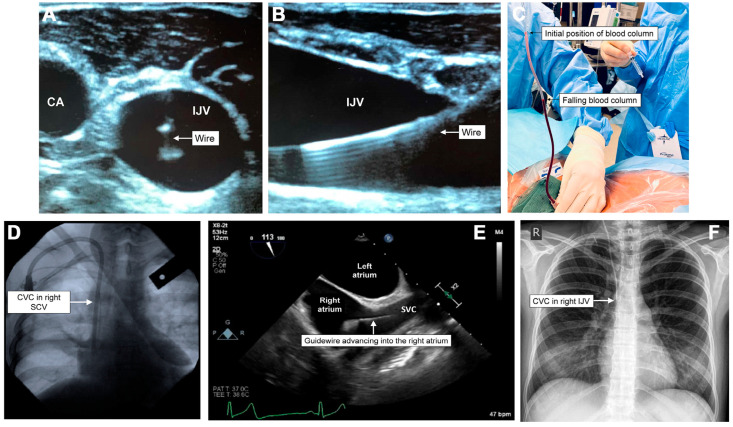
Confirmatory modalities for successful central venous cannulation, including dynamic ultrasound in short-axis (**A**) and longitudinal (**B**) views; manometry (**C**); fluoroscopy (**D**); transesophageal echocardiography (**E**); and chest radiography (**F**) (with permission from Walsh and Fitzsimons, 2023) [20].

**Figure 3 healthcare-13-01168-f003:**
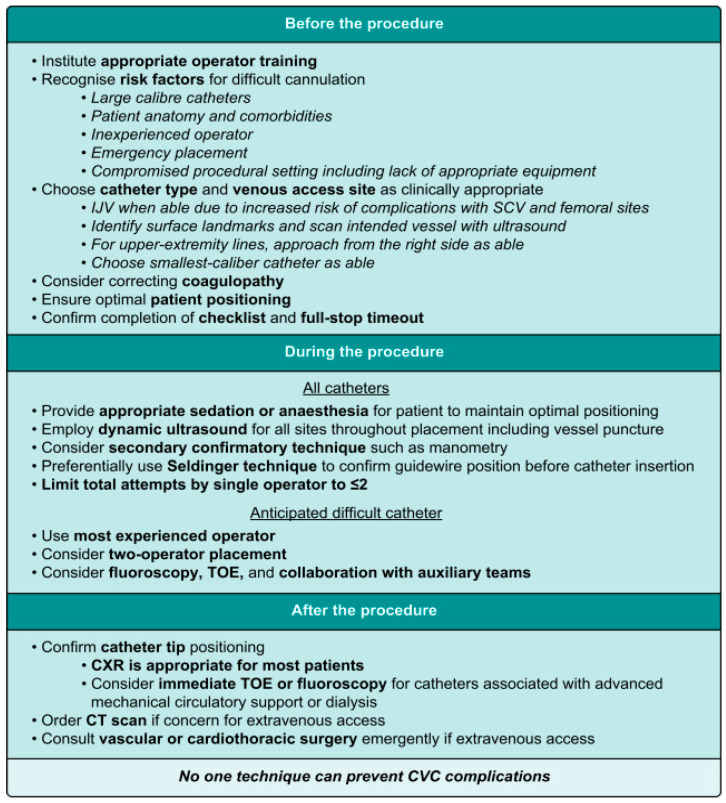
Suggested procedural approach to prevent CVC insertion complications (with permission of Walsh and Fitzsimons, 2023) [20].

**Figure 4 healthcare-13-01168-f004:**
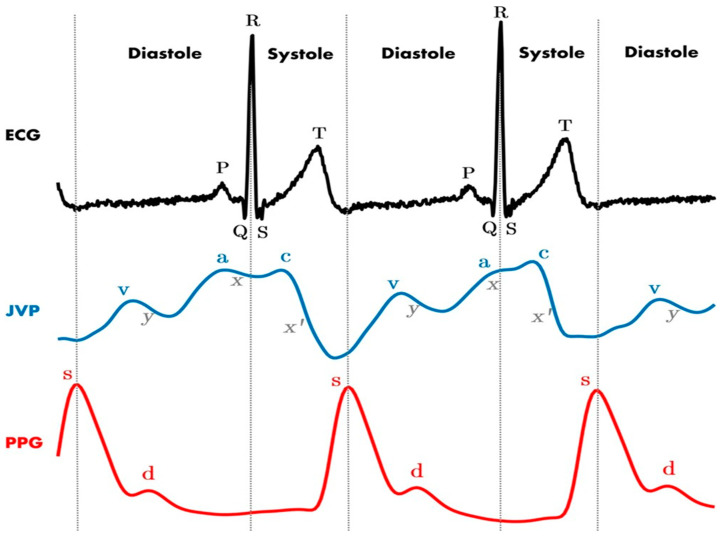
The jugular venous pressure waveform (JVP) compared to the arterial pressure waveform (carotid pulse on photoplethysmography—PPG). Recognizing the different waveforms on CVC helps proceduralists determine when an inadvertent arterial puncture occurs. For ECG, *P* corresponds to atrial depolarization, *QRS* complex to ventricular depolarization, and *T* to ventricular repolarization. For JVP: The (a) wave represents atrial contraction and is followed by the (x) descent, indicating the closure of the tricuspid valve. Subsequently, the (c) wave shows the bulging of the tricuspid valve due to right ventricular contraction, followed by the drop in pressure (x’). The (v) wave reflects the maximum pressure from right atrium filling before the tricuspid valve reopens. Finally, there is a drop in pressure during rapid ventricular filling, represented by the (y) wave. For PPG: *s* indicates the systolic pressure peak and *d* the diastolic pressure peak. (with permission of García-López and Rodriguez-Villegas, 2020) [53].

**Figure 5 healthcare-13-01168-f005:**
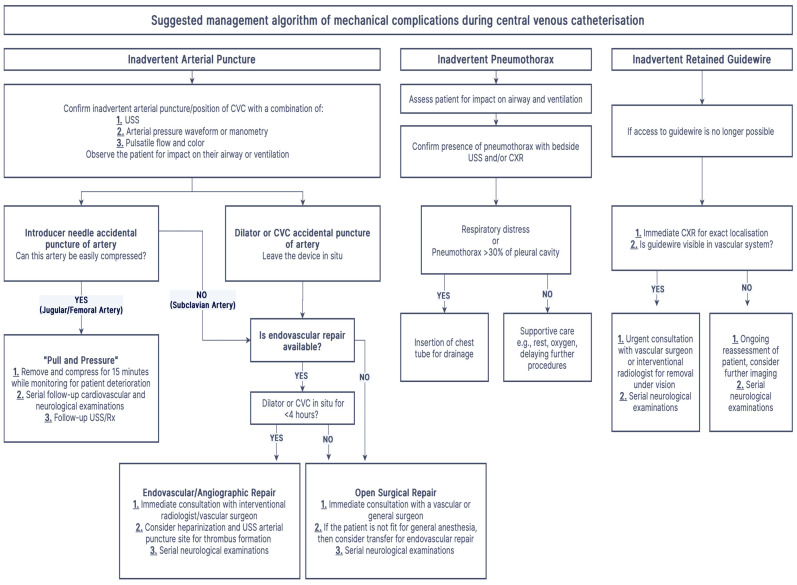
Suggested management algorithm of mechanical complications during CVC insertion.

## Data Availability

The original contributions presented in this study are included in the article. Further inquiries can be directed to the corresponding author.

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
