# Peer review of "Central Venous Access: An Update on Modern Techniques to Avoid Complications"

_healthcare, 2025, doi:10.3390/healthcare13101168_

Round 1
Reviewer 1 Report
Comments and Suggestions for Authors
This manuscript presents a comprehensive and well-articulated narrative review of central venous catheterization (CVC), from historical developments to modern techniques, complications, and their management. The topic is timely and highly relevant for anesthesia and critical care clinicians. The review is thoroughly referenced, drawing from high-quality evidence and international guidelines, and offers several practical insights for trainees and experienced practitioners.
I commend the authors for their clear structure and clinical emphasis. However, several minor revisions would improve the manuscript's clarity, precision, and utility.
Major Comments
- Lack of Methodology Description
Although this is a narrative review, it would benefit from a brief section outlining the literature review process (e.g., databases searched, inclusion criteria, date ranges). This would enhance transparency and reproducibility.
Suggest placing this after the introduction or in a brief "Methods" subsection.
- Repetition of Concepts
Sections repeatedly describe the use of ultrasound guidance and its benefits. Consolidating this information would streamline the narrative and improve flow.
- Speculative Future Directions
The discussion on obesity and new complications is thought-provoking but speculative. If available, consider referencing preliminary data or rephrasing with greater caution.
Suggest including a call for prospective research to explore these hypotheses.
Minor Comments
- Terminology and Consistency
Standardize terminology: consistently use "ultrasound" or "USS" throughout the manuscript.
Replace some overly informal phrases (e.g., "look into the future") with more formal academic language.
- Figures and Tables Integration
Ensure all figures and tables are explicitly referenced and discussed in the body of the text at appropriate points.
Table 6 could benefit from additional clinical context (e.g., operator experience, setting, sedation status).
- Grammar and Style
Minor grammatical errors (e.g., missing commas, occasional run-on sentences) are present and should be corrected for improved readability.
Sentence structure in several places (especially transitions between sections) can be tightened.
- Figure Legends
Ensure that figure legends include enough context to be understandable on their own. For example, Figure 4 should explicitly state the relevance of waveform differences to catheter placement safety.
Reviewer 2 Report
Comments and Suggestions for Authors
The paper is well-written and informative. However, for a narrative review, it is recommended to include a short methodology section describing how the literature was searched (databases used, keywords applied) and possibly a flowchart of literature outlining complications related to central venous access.
The focus appears to be on traditional methods of central venous access—jugular, subclavian, and femoral. I suggest to broaden the scope to include contemporary techniques for accessing the central venous system, such as peripherally inserted central catheters (PICC) and port-a-caths. These methods are solutions for long-term central venous access. PICC lines are frequently used in different settings, while port-a-caths are another variation of central veins access with specific placement procedures and associated complications. Including these modern techniques will improve the paper, in my opinion.
Reviewer 3 Report
Comments and Suggestions for Authors
Thank you very much for the opportunity to review the manuscript entitled “Central venous access: an update on modern techniques to avoid complications”. While reviewing the article, I had the following comments and questions:
LL45: Which complications were preventing the mainstream use?
The conception of this narrative review needs modification. Aspects mentioned in chapter 2 about the procedure take for granted that the reader is familiar with the detailed and site-specific complications.
Rather the (site-specific) complications should be thematized early in this Review to ease understanding of suggestive statements articulated in the procedure chapter.
Many introductions of the different chapters are based on findings of the systematic review and meta-analysis of reference number 2.
LL116 ff: The carina is another reliable landmark indicating positioning of the CVC outside the pericardium and visible even in CXRs (e.g. Schuster M et al.: Br J Anaesth 2000; 85: 192 – 4) and should be mentioned/discussed here.
LL 157 – 160 References missing.
LL 161 – 163 This first statement of gold standard somehow contradicts the statements of chapter 2 where alternatives of CXR are the preferred method. This paragraph also indicates the cross linkage between the chapters because it duplicates the topic. It might give the impression that the chapters were written independently and fused later but do not really match.
LL 175 Reference is missing
Moreover, the study demonstrated different predominant complications for different insertion sites. Results should be reported more differentiated because this large study among others might impact critical thinking about the optimal insertion site in patient specific conditions.
Table 3 – What references are data based on that are reported in Table 3?
Same with Table 4
L 222 – Reference needed for the statement as “most dangerous and difficult-to-manage”
L 233 – Reference needed.
LL 264 – 265 Is this a finding/conclusion of this review?
LL 305 ff: Should it not be all three, the intensivist, the interventional radiologist and the vascular surgeon discussing in a shared decision making process which is the optimal therapeutic approach balancing the patient-individual and procedure-individual risks and benefits?
LL 320 – 324 contradictory – if no angiography available consider open surgical repair under general anesthesia, but if not stable enough for general anesthesia consider angiography?
General anesthesia can be required for open surgical repair as well as interventional radiology. Moreover, also open surgical repair might be done under local or regional anesthesia, the need for general anesthesia should not be a factor for decision. In general, no one is too unstable for general anesthesia. Moreover, sick patients are often too unstable for anything else but general anesthesia.
LL 326 – 329 are pneumothorax rates for CVCs inserted in the femoral veins as high? How many of the 1000 CVCs wee inserted in the femoral veins?
LL 331 ff – is immediate CXR after CVC insertion recommended? Or do recommendations recommend a certain time interval until the CXR should be taken the earliest?
LL 339 ff: drainage of a pneumothorax only to prevent a tension pneumothorax?
LL 335 / Table 7 – What about other CVC-associated pleural effusions (e.g. chylothorax, infusothorax) which are rare but known complications that require differentiated diagnosis and treatment.
L 385 Reference strongly needed
LL 394 – Catheter-associated infections are a a frequent complication and a relevant factor in infection management in the ICU. Additional elaboration on this topic should be done including a differentiated discussion on catheter removal or alternative treatment options and antiinfective management.
Round 2
Reviewer 3 Report
Comments and Suggestions for Authors
The authors sufficiently answered all questions and comments.
Few typos and grammar errors should be erased before publication.
Author Response
Comment:
The authors sufficiently answered all questions and comments.
Few typos and grammar errors should be erased before publication.
Response:
Thank you for your help refining our article. We agree with your comment. Therefore, we have gone through the article and attempted to fix a variety of small grammatical errors and ensure that it is written entirely in consistent American English before resubmission for publication.